# Use of Riluzole for the Treatment of Hereditary Ataxias: A Systematic Review

**DOI:** 10.3390/brainsci12081040

**Published:** 2022-08-05

**Authors:** Iván Nicolas Ayala, Syed Aziz, Jennifer M. Argudo, Mario Yepez, Mikaela Camacho, Diego Ojeda, Alex S. Aguirre, Sebastian Oña, Andres F. Andrade, Ananya Vasudhar, Juan A. Moncayo, Gashaw Hassen, Juan Fernando Ortiz, Willian Tambo

**Affiliations:** 1KER Unit, Mayo Clinic, Rochester, MN 55902, USA; 2Shaheed Suhrawardy Medical College, Dhaka 1207, Bangladesh; 3School of Medicine, Colegio de Ciencias de la Salud, Universidad de Cuenca, Cuenca 010107, Ecuador; 4School of Medicine, Colegio de Ciencias de la Salud, Universidad Católica Santiago de Guayaquil, Guayaquil 090615, Ecuador; 5School of Medicine, Colegio de Ciencias de la Salud, Universidad San Francisco de Quito, Quito 170901, Ecuador; 6Aster RV Hospital, Bengaluru 560078, Karnataka, India; 7School of Medicine, Pontificia Universidad Católica del Ecuador, Quito 17012184, Ecuador; 8Capital Region Medical Center, University of Maryland, Largo, MD 20774, USA; 9Neurology Department, Spectrum Health/Michigan State University, Grand Rapids, MI 49503, USA; 10Feinstein Institute, Northwell Health, New York, NY 11030, USA

**Keywords:** spinocerebellar ataxia, riluzole, NMDA receptor antagonist

## Abstract

Ataxia is a constellation of symptoms that involves a lack of coordination, imbalance, and difficulty walking. Hereditary ataxia occurs when a person is born with defective genes, and this degenerative disorder may progress for several years. There is no effective cure for ataxia, so we need to search for new treatments. Recently, interest in riluzole in the treatment of ataxia has emerged. We conducted this systematic review to analyze the safety and efficacy of riluzole for treating hereditary ataxia in recent clinical trials. We conducted a systematic review using PubMed and Google Scholar as databases in search of this relationship. We used the Preferred Reporting Items for Systematic Reviews and Meta-Analyses (PRISMA) and Meta-analysis of Observational Studies in Epidemiology (MOOSE) protocols to conduct this study. For inclusion criteria, we included full-text clinical trials on humans written in English and found three clinical trials. We excluded case reports, literature reviews, systematic reviews, and meta-analyses for this analysis. We aimed to evaluate the Scale for the Assessment and Rating of Ataxia (SARA) score, the International Cooperative Ataxia Rating Scale (ICARS) score, and the safety of the medication. Two out of the three clinical trials showed statistically significant clinical improvement in the ICARS and SARA scores, while the other trial did not show improvement in the clinical or radiological outcomes. The drug was safe in all clinical trials. Overall, the results of this analysis of riluzole for the treatment of hereditary ataxia are encouraging. Further clinical trials are needed to investigate the efficacy of riluzole on hereditary ataxia.

## 1. Introduction

Riluzole is a neuroprotective agent, initially approved by the Food and Drug Administration in 1996 to treat amyotrophic lateral sclerosis (ALS); riluzole is an N-methyl-D-aspartate (NMDA) receptor antagonist [1,2]. The neuroprotective properties of riluzole seem to reduce extracellular glutamate concentrations by inhibiting glutamate release [2]. This effect is attributed to the suppression of presynaptic calcium influx. The additional properties of riluzole include: the inhibition of persistent ion channels, inward currents, protein kinases, and the enhancement of intracellular heat shock proteins and domain potassium channels (TREK-1), whose roles are supposed to be neuroprotective [2]. Studies on riluzole have an uncovered the mechanisms that control the excitability of central and peripheral neurons.

Ataxia is a physical finding usually related to cerebellar dysfunction. However, abnormal sensory inputs into the cerebellum, such as dorsal column disease, can also result in ataxia [3]. The etiologies of cerebellar ataxias are diverse and can involve infectious and immune-mediated, genetic, and degenerative factors. Assessing gait imbalance is usually the first approach to recognizing ataxia; patients often describe difficulty going upstairs or downstairs, difficulty running, or trouble walking barefoot [3]. Patients also present eye movement abnormalities such as horizontal or vertical end-gaze nystagmus. Abnormal hand examination findings such as dysmetria or dysdiadochokinesia are usually present in ataxic patients. Imaging diagnostic tests such as magnetic resonance imaging (MRI) can provide a variable degree of cerebellar atrophy, where vermal atrophy might be associated with truncal and gait ataxia. In contrast, paravermal atrophy might be related to limb ataxia [3].

There have been three clinical trials involving riluzole in the treatment of hereditary ataxia in the following conditions: Friedreich’s ataxia (FRDA), spinocerebellar ataxia (SCA), multiple system atrophy (MSA), and GAD-related neurological syndrome. Table 1 shows the main features of these hereditary ataxias [4,5,6,7].

We conducted this systematic review to investigate riluzole’s efficacy in hereditary ataxias. We aimed to investigate whether riluzole improves the functional outcome in ataxia through SARA and ICARS scores, and if the medication is well-tolerated among these patients.

## 2. Materials and Methods

### 2.1. Protocol

We carried out a systematic review using the Preferred Reporting Items for Systematic Reviews (PRISMA) and Meta-analysis of Observational Studies in Epidemiology (MOOSE) protocols (Figure 1) [8,9].

#### 2.1.1. Eligibility Criteria and Study Selection

The inclusion criteria were: full-text observational and clinical trials conducted on humans and written in English. The exclusion criteria were: literature reviews, systematic reviews, meta-analyses, and case reports.

After screening the studies, we only included papers with one of the following characteristics: (1) patients with hereditary ataxias; (2) intervention: riluzole; (3) comparator: placebo or control; and (4) outcomes: Scale for the Assessment and Rating of Ataxia (SARA) score, the International Cooperative Ataxia Rating Scale (ICARS) score, and safety (common and severe adverse effects).

#### 2.1.2. Database and Search Strategy

We used the PubMed database for this systematic review. The search was conducted between 24 March 2022 and 31 March 2022. The search terms were: (“Riluzole”[Title/Abstract] AND “spinocerebellar ataxia”[Title/Abstract]) OR (“Riluzole”[Title/Abstract] AND “hereditary ataxia”[Title/Abstract]) OR (“Riluzole”[Title/Abstract] AND “Friedreich’s ataxia”[Title/Abstract]) OR (“Riluzole”[Title/Abstract] AND “ataxia”[Title/Abstract]).

#### 2.1.3. Data Extraction and Analysis

We collected the following information: author, year, country, SARA score (if available), ICARS score (if available), adverse effects, severe adverse effects, and conclusion of each clinical trial.

#### 2.1.4. Bias Assessment

We used the Cochrane collaboration risk of bias tool to assess the bias encountered in each study [10].

## 3. Results

On Figure 1 it is showed the clinical trials of this systematic review.

### 3.1. Study Characteristics

We found three clinical trials that specifically discussed the role of riluzole in the treatment of ataxia. On Table 2, showed the clinical trials of this systematic review [11,12,13]. 

Table 3 shows the outcomes and conclusions of the clinical trials of this systematic review [11,12,13].

### 3.2. Study Limitations

The limitations of the study by Ristori et al., 2010 are as follows: (a) short study period; (b) limited number of patients; (c) the heterogeneity of cerebellar ataxia was included; (d) the scale of assessment was ICARS (longitudinal) instead of SARA (linear); (e) while assessing patients, psychological state and fatigue were not considered, and inherently subjective components were included [12].

The limitations of the study by Romano, S. et al., 2015 are as follows: (a) although there were no compliance issues, for some patients, the protocol assessment was stressful, due to which non-informative results occurred occasionally; (b) only one subgroup of patients was given the SF-36 questionnaire; (c) with respect to randomization, there was a high loss of participating patients [13].

The limitations of the study by Coarelli, G. et al., 2022 are as follows: (a) an absence of neurofilament light-chain blood measurement, which better correlates with clinical and radiological outcomes; (b) an absence of objective biomarkers such as oculomotor recording or wearable sensors that might have provided more accurate readouts; (c) the CCAS scale was not used for cognitive assessment, which would have been helpful for the correlation of specific cerebellum lobule volumes; (d) a small number of enrolled patients in the study (45) [11]. Figure 2 shows the BIAS analysis of this study.

## 4. Discussion

### 4.1. Rationale of Riluzole for Hereditary Ataxia

In the study by Schmidt et al., mice treated with riluzole for ten months had decreased levels of the soluble ataxin-3 but an increase in ataxin-3 positive accumulation in the cortical brain regions. This suggests that riluzole did not alter the expression of the ataxin-3 protein. There was no improvement in motor deficits, behavior, or bodyweight in the mouse models [15]. PrP/SCA3 mice treated with riluzole presented with a more significant amount of damaged Purkinje cells. It was hypothesized that riluzole treatment paradoxically enhanced glutamate release in ATXN3-expressing cells, leading to an increased Ca^2+^ influx, causing ataxin-3 positive protein accumulation in Bergmann glial cells, and resulting in Purkinje cell damage [15]. Clinically, riluzole did not improve symptoms and was not beneficial for the mice in the long term. The author theorized that the riluzole treatment enhanced the glutamate release in the ATXN3-expressing cells, leading to an increased Ca^2+^ influx and resulting in Purkinje cell damage [15].

In the study by Shourmasti et al., mice were induced to tremor and ataxia with harmaline [16]. According to the excitotoxic hypothesis, harmaline causes the excessive release of glutamate in Purkinje cells, leading to excitotoxic damage. In a dose-dependent fashion, the duration and intensity of the tremors decreased in the treatment group compared with the placebo group, with statistically significant results [16]. Riluzole might have played a neuroprotective function in the harmaline-induced vermis Purkinje cell loss by the action of its inhibitory properties on glutamatergic neurotransmission [16].

The study by Janahmadi et al. used the neurotoxin 3-acetylpyridine (3-AP) to cause the degeneration of the inferior olive in rats through the climbing fibers (CFs) that innervate the Purkinje neurons of the cerebellum, simulating cerebellar ataxia. The administration of riluzole in these rat models almost completely protected the cerebellar Purkinje cells from degeneration, halted the development of ataxia, and partially improved motor behavior [17]. Histologically, the riluzole-treated rats had a normal Purkinje cell morphology compared to the scattered, irregular, and densely stained somas in 3-AP-treated rats [17]. The riluzole- and 3-AP-treated rats did not present intention tremors or signs of gait abnormalities in the 3-AP-only group. The mechanism of riluzole neuroprotection is not fully understood, but the findings suggest that the Ca2+-dependent K+ channel might play a role [17].

In summary, two out of the three animal/in vivo trials showed the efficacy of riluzole in the treatment of hereditary ataxias, while the last one showed promising results. The initial findings in animal studies justify the continuation of trials on human subjects. Figure 3 shows the main findings of the animal/in vivo studies [15,16,17].

### 4.2. Clinical Trials of Riluzole for Hereditary Ataxia

In the study by Coarelli et al., a dose of 50 mg of riluzole (Table 1) did not improve the clinical outcomes (SARA score) or the radiological features of patients with SCA 2. However, some promising findings could prompt future clinical trials. The drug was well-tolerated, with no serious adverse events and similar adverse events in both groups. The change in the SARA score did not vary after 12 months in either of the groups [11]. Velázquez-Pérez et al. commented on Coarelli et al.’s work that the study includes the most significant portion of patients with SCA [18]. Velázquez-Pérez et al. point out that the lack of efficacy in the trial may be due to the possible bias of using the one-point decrease in the SARA score as a cutoff for therapeutic success, because this parameter falls within the margin of error of the annual change score in patients with SCA 2 [18].

They also stress the curious worsening of the composite cerebellar functional Severity (CCFS) in the riluzole-treated group, mentioning the need to find more sensitive biomarkers to examine subtle motor changes. However, the possibility of the riluzole worsening the cerebellar ataxia was mentioned. There is uncertainty as to whether this was due to the study’s limitations or the lack of pathophysiological mechanisms of the drug [18]. The author suggests that the lack of efficacy of the drug could be related to the following reasons: (a) the drug has low cerebellar engagement; (b) low blood–brain permeability; (c) the inclusion of patients with marked Purkinje cell atrophy (maybe patients in early phases could benefit from the drug). Another critique is that Coarelli et al. did not include standardized physiotherapy in the study, which could have led to an effect modification bias [18].

In the study by Romano et al., a dose of 50 mg of riluzole improved the clinical outcomes (SARA score) of patients with Spinocerebellar Ataxia and FRDA (2:1 ratio) (Table 2). The results were statistically significant. The drug was well-tolerated, with no severe adverse event reports and a similar rate of adverse events between the treatment group and the placebo group. There was a steady clinical state in the riluzole arm that persisted for 12 months compared to the placebo arm, which showed deterioration at the end of the trial. The mechanism of action of riluzole in cerebellar ataxia is not well understood. While action in the small-conductance potassium channel openers seems plausible, the authors suggest a pleiotropic effect due to astrocyte’s enhanced uptake of glutamate and reduced glutamate release in the synaptic cleft counteracting the damage produced by excitotoxicity in the long term in cerebellar degeneration [12].

A comment on the Romano study by Bransma et al. mentioned that the observed effect was limited by interobserver and intraobserver variability when analyzing the most minor differences, consisting of at least a one-point reduction in the SARA score for improvement of ataxia in patients using riluzole; also, in a comment, Morales et al. suggested that a different trial should offer riluzole to patients initially allocated a placebo and test the SARA scores at three months to evaluate whether there is only symptomatic relief or if there is also a disease-modifying effect [19] However, Romano et al. replied that long-term disease progression studies demonstrated that there is an apparent yearly progression increase in the SARA score in different types of ataxias, most of them superior to one-point increases, suggesting that there is clinical significance to using a one-point reduction in the SARA score as a metric to assess clinical improvement [14].

In the Ristori et al. study, 50 mg of riluzole taken twice daily resulted in progressive clinical improvement (ICARS score) in patients with chronic cerebellar ataxia at the four-week and eight-week endpoints. The results were statistically significant in the ICARS score and the subscores for static function, kinetic function, and dysarthria. Overall, the drug was well-tolerated, with similar rates of adverse effects between the treatment group and the placebo group. Riluzole is an approved ALS treatment that opens small conductance Ca2+-activated potassium channels, which regulate the firing rate of neurons in deep cerebellar nuclei, reducing neuronal hyperexcitability. Thus, it has potential therapeutic benefits in the treatment of cerebellar ataxia. The study calls for further investigation into the mechanisms providing the positive outcome of riluzole on the heterogeneous patient population in their study (patients with a presumptive dentate nucleus and spinal pathology, patients with more selective cerebellar involvement, and patients with presumptive severe brainstem pathology) [12].

Two out of the three clinical trials showed statistically significant results, while the third study did not show clinical or radiological efficacy. Future studies should reduce the heterogeneity of the study population, as in Corelli et al.’s study focusing on a select group of hereditary ataxias. Additionally, neuroprotective drugs may work better in the early stages of the disease. For example, edaravone, another neuroprotective drug used in ALS treatment, initially failed to show clinical efficacy. However, clinical efficacy was found when the drug was used in the early stages of the disease [20].

## 5. New Perspectives (Future)

Troriluzole is a prodrug (third generation) of riluzole. It mainly acts by reducing glutamate levels at synapses, and it does so by increasing the glutamate uptake of glial cells by expressing more excitatory amino acid transporters (i.e., EAAT2). The drug is being studied in phase 2–3 clinical trials for the possible treatment of Alzheimer’s disease, treatment-resistant obsessive-compulsive disorder, and spinocerebellar ataxia.

Given the drug’s efficacy in animal studies and the promising results of clinical trials, exploring the effects of troriluzole on patients with hereditary ataxia might improve these patients’ symptoms in future studies.

Riluzole is currently being tested in clinical trials in different phases. Table 4 has a summary of the ongoing work that is available at clinicalstrials.gov (accessed on 5 May 2022) [21]. Table 4 shows the ongoing clinical trials of Ataxia and riluzole [21,22,23,24,25].

## 6. Conclusions

The efficacy of riluzole was shown in two of the three animal/in vivo trials on the treatment of hereditary ataxias, while the third one showed promising results. These initial findings in animal studies justify the continuation of trials on human subjects.

Two of the three clinical trials showed statistically significant results, whereas the third study did not show clinical or radiological efficacy. Future studies should reduce the heterogeneity of the study population, and a select group of hereditary ataxias should be used when conducting clinical trials. Additionally, neuroprotective drugs may work better in the early stages of the disease, and future studies could compare the drug’s effectiveness in patients with early- and late-onset varieties of the disease. For example, edaravone, another neuroprotective drug used in ALS treatment, initially failed to prove clinical efficacy.

However, clinical efficacy was found when the drug was used in the early stages of the disease.

There is a new drug on the market, troriluzole, which is being studied to treat ataxia. It is unclear if this drug will lead to meaningful clinical improvements in patients with ataxia.

## Figures and Tables

**Figure 1 brainsci-12-01040-f001:**
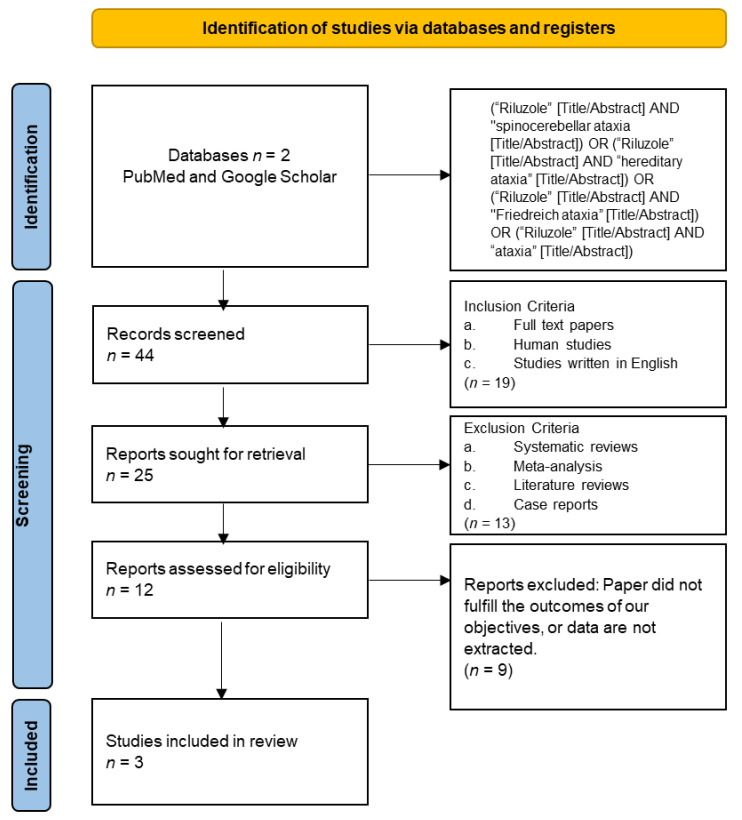
PRISMA flow chart used for this systematic review.

**Figure 2 brainsci-12-01040-f002:**
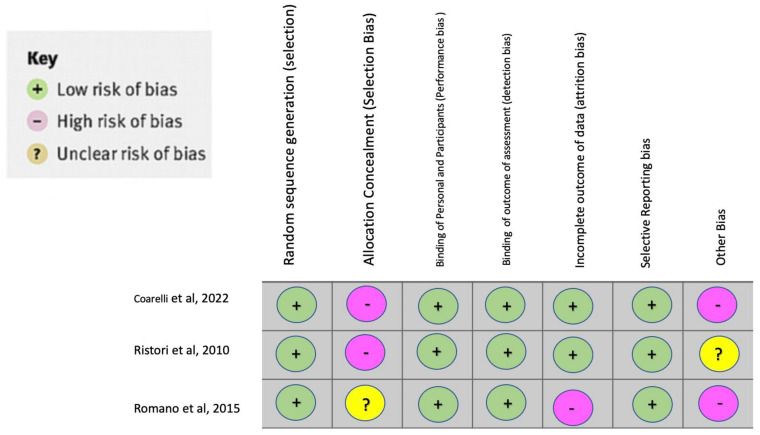
Bias Analysis.

**Figure 3 brainsci-12-01040-f003:**
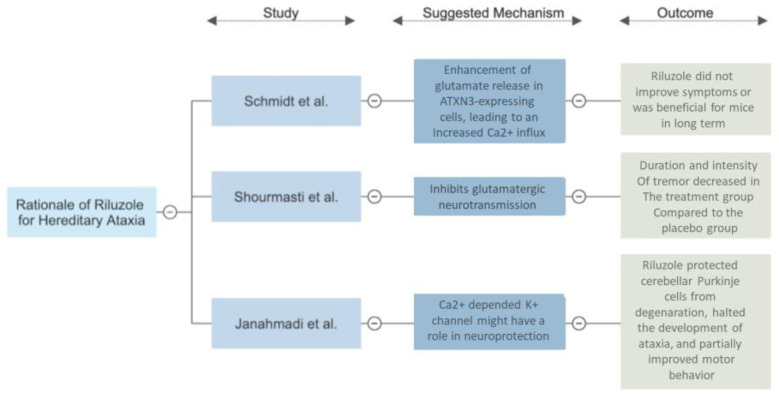
Rationale of Riluzole for Hereditary Ataxias; Comparison of the 3 studies included, suggested mechanism, and outcomes.

**Table 1 brainsci-12-01040-t001:** Main features of these hereditary ataxias.

Hereditary Ataxia	Clinical Features
Friedreich’s ataxia [4]	The most common disorder is FRDA, caused by a GAA trinucleotide repeat expansion in the Frataxin gene. FRDA causes progressive ataxia, dysarthria, cardiomyopathy, and an increased risk of diabetes mellitus; most patients require a wheelchair by the age of 15 with a marked reduction in lifespan, with the onset of death being around 36 years.
Spinocerebellar ataxia [5]	Transmitted in an autosomal dominant fashion; caused by CAG nucleotide repeat expansions that encode polyglutamine. The classic triad of SCAs includes gait ataxia and incoordination, nystagmus, and dysarthria; however, patients can present with additional findings such as pyramidal or cognitive dysfunction. In the last decade, much progress has been made by targeting downstream pathways with antisense oligonucleotides (ASOs), providing therapeutic relief in some patients.
GAD-related neurological syndrome [6]	GAD-related neurological syndrome, specifically cerebellar ataxia, is another rare hereditary autoimmune cerebellar disorder associated with genetic and environmental risk factors and the presence of GAD (glutamic acid decarboxylase) antibodies. The pathogenic role of GAD Ab in cerebellar ataxia is a reduced GABAergic transmission. The clinical presentation includes gait ataxia, dysarthria, and nystagmus, corresponding to primary participation of the cerebellar vermis. The disease shows a symptom onset that ranges from subacute to chronic, and the disease progression can last from months to years. Brain MRI reveals vermis atrophy, and CSF analysis shows CSF-specific oligoclonal bands and intrathecal synthesis of GAD Ab.
Multiple system atrophy—cerebellar phenotype [7]	MSA is part of the family of a-synucleinopathies, and there are two main types, the MSA parkinsonian type (MSA-P) and cerebellar (MSA-C). MSA-C type is caused by olivopontocerebellar atrophy degeneration. The main symptoms are gait and limb ataxia, dysarthria, eye movement abnormalities such as dysmetria, and saccadic intrusion. The diagnosis is mainly clinical, and treatment is only symptomatic.

**Table 2 brainsci-12-01040-t002:** The characteristics of clinical trials used in the systematic review.

Author and Year of Publication	Country	Study Design	No. of Pts. in the Treatment Group	No. of Pts. in the Control Group	Patient Selection	Dose, Duration, Route, of Administration
Coarelli et al., 2022 [11]	France	Clinical Trial	22	23	Patients with SCA Type 2	Riluzole 50 mg orally or placebo twice per day for 12 months
Romano et al., 2015 [13]	Italy	Clinical trial	28 (19 SCA; 9 FRDA)	27 (19 SCA; 8 FRDA)	Patients with SCA or FRDA	Riluzole 50 mg orally or placebo twice daily for 12 months
Ristori et al., 2010 [12]	Italy	Clinical trial	19	19	Patients with cerebellar ataxias of different etiologies: 8 FRDA, 8 SCA (type 1 (2), type 2 (4), type 2b (2)), 6 multiple system atrophy type C, 2 multiple sclerosis, 1 anti-GAD, 1 anti Yo, 13 ataxias of unknown origin	Riluzole 100 mg/day or placebo for 8 weeks

**Table 3 brainsci-12-01040-t003:** The outcomes and conclusions of the clinical trials of this systematic review.

Author and Year of Publication	Country	Outcome	Results	Main Conclusion
Coarelli et al., 2022 [11]	France	SARA score	SARA score improved 1 point in 7 patients (32%) in the treatment group versus 9 patients in the placebo group (39%), with a mean difference of −10.3% (95% CI −37.4% to 19.2%; *p* = (0.75). There was a median increase in the SARA score by 0.5 points (IQR −1.5 to 1.5) in the riluzole group versus a 0.3 point (−1.0 to 2.5) increase in the placebo group (*p* = 0.70). In both groups, the number of patients who experienced adverse events was similar (Riluzole, 16 (73%) patients vs. placebo, 19 (83%) patients; *p* = 0.49). The severity of these parameters was measured with the Common Terminology Criteria for Adverse Events.	There was no clinical or radiological improvement in patients with SCA type 2 treated with riluzole. There were no serious adverse events reported in the riluzole group. The adverse events were not statistically significant compared to the control group.
Romano et al., 2015 [14]	Italy	SARA score	In the riluzole group, 14 of 28 (50%) patients showed an improvement in SARA score compared to 3 (11%) of 27 patients in the placebo group (OR 8.00, 95% CI 1.95–32.83; *p* = 0.002). There were no severe adverse events recorded in either group.Two participants in the riluzole group had an increase in liver enzymes (less than two times above normal limits). Mild adverse effects were recorded in two of the riluzole participants and two of the placebo participants.	Riluzole improved clinical outcomes in patients with SCA and FRDA with statistically significant results. The medication was well tolerated.
Ristori et al., 2010 [12]	Italy	ICARS	A 5-point reduction in the ICARS was considered clinically relevant at the time of the trial’s design. The riluzole group presented a significantly higher ICARS 5-point drop compared to placebo after 4 weeks in 9/19 treatment group versus 1/19 in the placebo group. (OR) = 16.2; 95% confidence interval (CI) 1.8–147.1) and After 8 weeks the ICARS 5-point drop drop in 13/19 cases in the treatment group versus 1/19 in the place group; OR = 39.0; 95% CI 4.2–364.2). In the subgroups of the ICAR 5 score, this were the results: Total ICARS Score: After treatment, the mean change in the riluzole group was 7.05 (4.96) versus 0.16 (2.65); (*p* < 0.001) in the placebo group. Static sub scores: Decreased in the treatment group −2.11 (2.75) versus 0.68 (1.94); (*p* < 0.001) in the place group. For kinetic function in the riluzole group the score changed −4.11 (2.96) versus 0.37 (2.0); (*p* < 0.001) in the placebo group. For Dysarthria there was an improvement in the function in the riluzole group −0.74 (0.81) versus 0.05 (0.40) (*p* < 0.001) in the placebo. Adverse effects were sporadic and minor. *Values are mean (SD).*	Riluzole was effective as symptomatic therapy in various disorders that cause cerebellar ataxia. The results were statically significant. There were no major side effects in either group, or the treatment was well tolerated.

ICARS: International Cooperative Ataxia Rating Scale; Scale for the Assessment and Rating of Ataxia: SARA. SCA: Spinocerebellar ataxia, FRDA: Freidreich Ataxia.

**Table 4 brainsci-12-01040-t004:** Ongoing clinical trials of Ataxia and riluzole.

Author and Year of Study Start Date	Study Name	Conditions treated	Primary Outcomes	Secondary Outcomes	Dose, Duration, Route, of Administration
Ristoroi G. 2018 [21].	Riluzole in Patients with Spinocerebellar Ataxia Type 7. A Randomized, Double-blind, Placebo-controlled Pilot Trial with a Lead in Phase	Spinocerebellar Ataxia Type 7	1. Visual acuity expressed as log MAR units 2. Proportion of patients with stable SARA score	1. Farnsworth D15 Arrangement Test 2. Visual evoked potentials 3. Electroretinography 4. Optical Coherence tomography 5. SARA score	Treated: Riluzole 50 mg Orally twice daily for 12 months Control: Placebo drug for 6 months and riluzole during the last 6 months of study
Perlman S. 2018 [22]	An Open Pilot Trial of BHV-4157	SCA, SCA Type 1, SCA Type 2, SCA Type 3, SCA Type 6, and MSA-C	SARA Score	1. 8-Meter Walk Test 2. Sheehan Suicidality Tracking Scale 3. Beck Depression Inventory 4. Beck Anxiety Inventory	BHV-4157 dose, duration, and route not specified
Ristoroi G. 2010 [23]	Efficacy of Riluzole in Hereditary Cerebellar Ataxia	Cerebellar Ataxia	SARA Score	1. Baropodometric parameters 2. Quality of life: SF-36 3. Depression: Beck Scale	Riluzole 50 mg orally or placebo twice per day for 12 months
Durr A. 2018 [24]	Multicenter, Randomized, Double Blind, Placebo Controlled Clinical Trial with Riluzole in Spinocerebellar Ataxia Type 2	Spinocerebellar Ataxia Type 2	Change in Ataxia symptoms SARA score	1. Change in Ataxia symptoms SARA Score 2. Change in extracerebellar symptoms (Inventory of Non-Ataxia Signs (INAS)) 3. 12 months survival	Riluzole 50 mg orally or placebo twice per day for 12 months
Ristoroi G. 2005 [25]	Phase 2 Study of Riluzole Effects on Patients with Chronic Cerebellar Ataxia	Cerebellar Ataxia Hereditary Ataxia Multiple Sclerosis	ICARS total scores and subscores (oculomotor, kinetic, postural, speech)	None	Riluzole 50 mg orally or placebo twice per day for 8 weeks

## Data Availability

Not applicable.

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
