# Peer review of "Use of Riluzole for the Treatment of Hereditary Ataxias: A Systematic Review"

_brainsci, 2022, doi:10.3390/brainsci12081040_

Round 1

Reviewer 1 Report

Ayala et al. presented in the review an overview about three published clinical studies of riluzol in ataxia and also discussing potential pro and con´s of these studies. The manuscript is well written and easy to follow. For people who are interested in the filed, this review presents an easy to follow and very nice and good overview.

Some minor points:

(I) It would be helpful, if the author include also some information about the studies of riluzol in ataxia listed in ClinicalTrials.Gov. There are listed 5 for ataxia (3 completed), 1 completed for FRDA, 2 for MDA. It would be helpful if the author can include this information in the discussion, that there are also studies running currently and in the outlook, conclusion. That will give a more enhanced ovierview about studies finalized and running in the field.

(II) Study from Romano and Ristori included several different SCAs, but it is not mentioned in the review which SCAs were included with how many participnats of each group. To have a full overview in the review that need to be clarified

(III) In the study limitation, it was mentioned for the study from Coralli, that they did not analysed NFL and that this is limiting the study. But it was not mentioned for the other studies. Did that mean that all other studies included NFL as outcome measure? It would be helpful if the author can demonstrate in a table all outcome parameter from all three studies. This would make it easier why Ayale et al discussed different parameters as weakness of a specific study.

(IV) There are two tables labbeled with table 1: (A) describtion and features of ataxias included in the review and (B) characteristics of clinical trials discussed in the review

(V) In table 1 (clincial characteristics) the study country for Coarelli et al is Italy, but in table 2 the study country for Coarelli is France. What is correct?

(VI) There are several spelling errors, e.g. table 1 - clicnical features of hereditary ataxia - Frederick Ataxia instead of Friedreich Ataxia; line 129 "tem months" instead of "ten months"; line 229 "the dug is" instead of "the drug is"; line 250 "In unclear of the moment"

(VII) At line 160 and line 236 the author mention "two animal/ in vivo trials" but they are describing three shortly before.

Author Response

(I) It would be helpful, if the author include also some information about the studies of riluzol in ataxia listed in ClinicalTrials.Gov. There are listed 5 for ataxia (3 completed), 1 completed for FRDA, 2 for MDA. It would be helpful if the author can include this information in the discussion, that there are also studies running currently and in the outlook, conclusion. That will give a more enhanced ovierview about studies finalized and running in the field.

1. We have reviewed the future clinical trials and added a short paragraph in section 5, and also added a table.

(II) Study from Romano and Ristori included several different SCAs, but it is not mentioned in the review which SCAs were included with how many participnats of each group. To have a full overview in the review that need to be clarified

2. We have enumurated the types of SCA in the studies, and the number of participants. This is enuntiated in table 2

(III) In the study limitation, it was mentioned for the study from Coralli, that they did not analysed NFL and that this is limiting the study. But it was not mentioned for the other studies. Did that mean that all other studies included NFL as outcome measure? It would be helpful if the author can demonstrate in a table all outcome parameter from all three studies. This would make it easier why Ayale et al discussed different parameters as weakness of a specific study.

3. In table 3 we added a column of the outcomes of each of the studies, so the reader has clearer picture of the study.

(IV) There are two tables labbeled with table 1: (A) describtion and features of ataxias included in the review and (B) characteristics of clinical trials discussed in the review

4. We corrected the mistake as suggested.

(V) In table 1 (clincial characteristics) the study country for Coarelli et al is Italy, but in table 2 the study country for Coarelli is France. What is correct?

5. The authors are from Italy, but study was conducted in France.

(VI) There are several spelling errors, e.g. table 1 - clicnical features of hereditary ataxia - Frederick Ataxia instead of Friedreich Ataxia; line 129 "tem months" instead of "ten months"; line 229 "the dug is" instead of "the drug is"; line 250 "In unclear of the moment"

6. We corrected the mistake as suggested.

(VII) At line 160 and line 236 the author mention "two animal/ in vivo trials" but they are describing three shortly before

7. In summary, two out of three studies showed the efficacy of riluzole in the treatment of Hereditary ataxias, while the last one showed promising results. Nevertheless these are only animal/in vivo studies.

We also add all the suggestion made by the other reviewer, the major change is adding a graphic in section 4.1. I think this gives the reader a bettter understanding of the animal/in vivo studies.

Thank dear reviewer, I hope we have solved all your queries.

Kind Regards

Fernando Ortiz

Reviewer 2 Report

Ayala et al. summarized three clinical trials using riluzole for the treatment of hereditary ataxias, commented on the clinical trial designs and outcomes, and made suggestions for future trials. Here are my comments.

  1. Figure 1 is not very much relevant to the scientific analyses of the clinical trials. I would suggest moving it to the supplementary.
  2. The mechanistic studies discussed in Section 4.1 are not conclusive. A diagram summarizing current findings will be helpful.
  3. In Section 5, the authors suggested one different drug that has the potential to improve the patients’ symptoms. It’s not clear if the authors think the new one will be better than riluzole, which is the topic of the review. In addition, it’s not mentioned if there are other potential treatments.
  4. Coarelli et al, 2022 was labeled in different countries in Table 1 and 2.
  5. There are typos and unfamiliar usage of expressions throughout the manuscript. Language editing is strongly suggested.

Author Response

1. Figure 1 is not very much relevant to the scientific analyses of the clinical trials. I would suggest moving it to the supplementary.

Dear reviewer, are you referring to the PRISMA flow chart, we thought it was obligatory to include it, please let us have your opinion, and we will address it again.

2. The mechanistic studies discussed in Section 4.1 are not conclusive. A diagram summarizing current findings will be helpful.

A Graphic was added.

3. In Section 5, the authors suggested one different drug that has the potential to improve the patients’ symptoms. It’s not clear if the authors think the new one will be better than riluzole, which is the topic of the review. In addition, it’s not mentioned if there are other potential treatments.

3. Coarelli et al, 2022 were labeled in different countries in Tables 1 and 2.

The change was made, is a label in France in both studies.

5. There are typos and unfamiliar usage of expressions throughout the manuscript. Language editing is strongly suggested.

Grammar typos were corrected by two different authors independently.

Dear Reviewer thanks for reviewing the paper, we appreciated all your suggestions, and we hope, we have addressed them correctly.

Kind Regards,

Fernando Ortiz.

Round 2

Reviewer 2 Report

Thank you for addressing my concerns.

Author Response

We adress minor changes suggested by the editor